# Representations of Psychoactive Drugs’ Use in Mass Culture and Their Impact on Audiences

**DOI:** 10.3390/ijerph18116000

**Published:** 2021-06-03

**Authors:** Marek A. Motyka, Ahmed Al-Imam

**Affiliations:** 1Institute of Sociological Sciences, University of Rzeszow, 35-310 Rzeszów, Poland; mmotyka@ur.edu.pl; 2Department of Anatomy and Cellular Biology, College of Medicine, University of Baghdad, Baghdad 10001, Iraq; 3Alumni Ambassador, Barts and the London School of Medicine and Dentistry, Queen Mary University of London, London E1 2AD, UK

**Keywords:** mass culture, alcohol, marijuana, illegal drugs, youth, COVID-19

## Abstract

Drug use has been increasing worldwide over recent decades. Apart from the determinants of drug initiation established in numerous studies, the authors wish to draw attention to other equally important factors, which may contribute to augmenting this phenomenon. The article aims to draw attention to the content of mass culture, especially representations of drug use in mass media, which may influence the liberalization of attitudes towards drugs and their use. The role of mass culture and its impact on the audience is discussed. It presents an overview of drug representations in the content of mass culture, e.g., in film, music, literature, and the occurrence of drug references in everyday products, e.g., food, clothes, and cosmetics. Attention was drawn to liberal attitudes of celebrities and their admissions to drug use, particularly to the impact of the presented positions on the attitudes of the audience, especially young people for whom musicians, actors, and celebrities are regarded as authorities. Indications for further preventive actions were also presented. Attention was drawn to the need to take appropriate action due to the time of the COVID-19 pandemic when many people staying at home (due to lockdown or quarantine) have the possibility of much more frequent contact with mass culture content, which may distort the image of drugs.

## 1. Introduction

Drug use data published in 2020 by the United Nations indicate that drug use is increasing worldwide, both in terms of the total number and percentage of the global drug-using population. Over the last decade (2009 to 2018), the estimated proportion of drug users increased from 210 to 269 million users (4.8% to 5.3% of the global population). In 2017, 585,000 deaths globally were attributed to drug use due to premature death related to consequences of use and overdose [1]. The European Monitoring Centre for Drugs and Drug Addiction (EMCDDA) highlighted the high availability of drugs, especially cannabis, hashish, cocaine, and heroin. EMCDDA found that in European Union countries, more than 8300 Europeans died from drug abuse during the survey year, with an average age of about 41 years. In addition, despite the preventive measures taken (e.g., the monitoring and early warning system in EMCDDA), the introduction of new synthetic opioids on the drug market is a worrying example of the continued adaptability of producers and traffickers of these drugs [2].

Predictors of drug initiation and use tend to be similar in different parts of the world. The most commonly reported are; low parental support, upbringing in a dysfunctional family (poverty, violence, addictions, crime, and educational inefficiency), peer influences, deficits in social problem-solving skills, early exposure to tobacco influencing the initiation of marijuana experimentation, promotion of casual attitudes towards drugs by pop culture idols, low economic status, low religiosity, weakened family relationships and lack of parental control, weak social support network, availability of drugs, and lack of knowledge about the dangers of their use [3,4,5,6,7].

It is highly likely that the increase in the use of cannabis, tranquilizers, and other drugs observed in the last year results from the need to reduce the stress associated with the coronavirus pandemic [8,9,10]. The pandemic not only forces millions of people into long-term confinement in their homes (due to lockdown, quarantine, and fear of infection) but, above all, increases tension and its long-term impact [11]. Due to the difficulty of using numerous natural methods to reduce tension (e.g., activity, work, and sport), many people try to seek solace in psychoactive drugs [9,12]. Studies indicate that alcohol consumption and other recreational drugs have increased significantly since the onset of the pandemic [13].

There may also be other reasons for the popularity of psychoactive drugs. In recent years, the use of snus (a smokeless tobacco product) has become increasingly common, and chewing it is sometimes advertised as an alternative to smoking to help smokers break their addiction [14]. The promotion of new substance use runs the risk of distorting the risks associated with its use, which, while they may be less, need to be communicated. However, intrusive advertising through various means, including sending free samples to potential customers, can influence the willingness to try these new, seemingly harmless substances [15]. Comparing agents with relatively milder psychoactive effects (e.g., snus, e-cigarettes, marijuana, and even energy drinks) with drugs that cause deep physical and psychological dependence (e.g., heroin, amphetamines, and cocaine) distorts the risks associated with the possibility of developing dependence even on so-called ‘light’ substances, and consequently the curiosity or need to reach for much potent and more destructive drugs [16,17].

However, attention needs to be paid to other conditions that have been present and perpetuated for decades around the world, to a greater or lesser extent favoring the use of psychoactive substances, not yet included in national prevention strategies or ignored due to the scarcity of research on the impact of drugs on drug-related attitudes or for other reasons that are difficult to identify, which is particularly important to introduce prevention strategies different from the ones used so far in the post-pandemic reality, both oriented to new circumstances and affecting hitherto underestimated risk factors [18]. These postulates concern, in particular, the need to implement educational strategies (e.g., building a positive self-image, conflict resolution, interpersonal skills, coping with stress, assertiveness, and learning to accept mass media content critically) and strategies of implementing activities alternative to drug use (e.g., developing passions, interests, and health promotion).

This article aims to draw attention to the role of mass culture content in the possibility of liberalizing attitudes towards psychoactive substances and their use in recipients of such messages.

## 2. Mass Culture and Its Impact

Mass culture is a specific form of symbolic culture, and an essential feature of its characteristic message is the absence of direct contact between the sender and the receiver of the transmitted content. Mass culture carriers are all means of mass media (including cinema, television, radio, advertising, and magazines), but above all, the relevant aspect is the content presented in them [19]. The popularity and dissemination of this type of message are fostered by technological development, especially the availability of the internet, enabling the diffusion of all ideas to an unlimited group of recipients. The relationships that occur between the participants of virtual social networks create new patterns of interaction. These networks offer a platform for exchanging views and information in which there are hierarchies of roles (administrators and users) and provide strong group membership [20].

There are many examples in the scientific literature of online social networks having a positive impact on the health attitudes of participants in such interactions [21,22,23]. Unfortunately, many examples have also been identified of internet groups disseminating views that may consequently foster harmful or risky behaviors, including promoting violence and the use of psychoactive drugs [24,25,26]. Analogous examples of the influence of cultural texts on audience attitudes can be observed in other settings, e.g., listening to music containing messages about the use of psychoactive drugs can influence alcohol abuse, drug use, and aggressive behavior of fans of such music [27]. Inaccurate representations of drug use portrayed in films may also distort the image of this phenomenon [28]. Some studies suggest a causal relationship between watching such films and the risk behaviors observed in viewers of these messages [29,30]. Researchers have indicated the need for extensive research on such topics [31].

Interactions between members of different social groups are constantly changing. Considering the dynamics of society, it can be assumed that the relationships between behavior and cultural products can also change. Norms, principles, patterns of behavior are constantly being modeled. Some views and attitudes are accepted depending on culturally created positions, while others are criticized and rejected. Authors of social studies indicate that the products of contemporary mass culture may play an essential role in the liberalization of attitudes towards psychoactive drugs, presenting drugs as a sanction-free element of social life, free of odium, available, and almost unlimited [32,33,34]. The normative changes taking place in the modern world concerning narcotic drugs, the creation of a positive image of them in numerous media, and at the same time, the weakening of the role of the family and secularization processes have been recognized as crucial determinants of the changes in attitudes towards psychoactive drugs observed today [6,35,36,37]. Reaching for drugs with psychoactive properties may also be conditioned by the attitudes towards such drugs acquired from the external environment; for example, the perception of drugs as safe, intelligence-enhancing, and at the same time, accessible [38].

Contemporary mass media exert a strong influence on the recipients’ behavior both in a planned and unplanned way. Educational actions carried out using mass communication media may be the source of accessible and reliable pro-health knowledge, and at the same time, they favor the modeling of behaviors expected by their senders [39]. At the same time, the same media may be a source of creating positive attitudes towards drugs and, as a result, drug use [40].

The scientific literature provides many theoretical concepts of the influence of mass media on the recipients of the exposed content, but we draw attention to two concepts in our considerations. The cultivation theory developed by George Gerbner and Larry Gross emphasizes the critical role of television as a medium reaching the widest audience, which by presenting a given social phenomenon often enough creates its perception among viewers. However, the reality shown on television is based more on speculation than facts [41]. According to Gerbner et al., the primary function of television is to perpetuate social patterns and cultivate resistance to change. The authors of the theory do not assume that frequent viewing of a given phenomenon will make us behave similarly to the characters of the watched broadcasts. However, frequent reception of certain behaviors cultivates and perpetuates the viewers’ consciousness of their occurrence in the social space [42]. This theory has been developed based on several research projects in which the social effects of violence presented in the media were determined. However, its assumptions have also been adapted to the research on the attitudes towards alcohol in the recipients of music videos whose musicians presented an accessible approach to this issue [43], and research on the attitudes and sexual behavior of students cultivated by the activities observed in the characters of popular TV series [44]. Adapting the assumptions of this theory to the study of the reality related to the portrayal of drugs in the media, it can also be assumed that the frequent exposure of content depicting psychoactive drugs, especially in a casual or humorous emotional way, may favor liberalization of attitudes towards these drugs.

Albert Bandura’s social learning theory also allows us to understand the modeling of media actors’ attitudes on the recipients of these messages. According to the author of this theory, reaching for drugs is a consequence of adopting similar behavior patterns observed in persons significant for a given individual. The social learning theory identifies the causes responsible for deviant behavior and allows for a better understanding of it. By observing the behavior of essential people in his or her environment, the individual acquires and models these behaviors in himself or herself. Models are part of the social transmission of culture and are an essential social learning element [45]. Drug use observed in significant others or liberal attitudes towards this type of behavior can also be adopted. The group of models whose behaviors are imitated by youth includes mainly close family members, influential people in the peer environment, and youth idols, e.g., music band leaders. Observation plays an essential role in social learning, during which the frequency of observed behavioral patterns favors a more substantial impact on an individual. Remembering the observed activities, reproducing them, and, as a result, adopting them as one’s own are the following stages of the learning process. Rewarding effects for accurate behavior imitation, including acceptance in a group, favor memorizing and faithful reproduction of observed behavior [7,45]. Strong associations with users of psychoactive substances significantly affect the imitation of this type of activity by individuals newly admitted to the community [46].

A tendency to act and behave similarly can be observed among recipients of mass culture content and a tendency to systematically become mass members of society who, imitating their idols, celebrities, and other people considered as authorities, duplicate the observed behaviors [47]. Technological progress, the emergence of new media, consumerism in its broadest sense, and incredibly persuasive and manipulative activities of the authors of mass-distributed content are among the many factors conducive to the standardization of messages and distortion of reality [48].

The cultivation and perpetuation of certain behaviors and the influence of broadly understood authority figures may be important factors favoring the spread of ideas liberalizing attitudes towards drugs.

## 3. Representations of Drug Use in Mass Culture

For several decades now, positive messages about psychoactive drugs have been frequent in mass culture productions. One can read about drugs or experiences related to their use in the press and books. There are films on the subject or films with drug themes. Music works on the subject, and internet portals created and maintained by people interested in drug-related issues almost freely present positions and experiences after drug use, as do some youth idols and celebrities [7]. In addition, representations of psychoactive drugs can be encountered while shopping, in grocery, clothing, drug or gadget shops, and on city streets, especially on buildings decorated with colorful pro-drug graffiti. Below is an overview of our observations.

### 3.1. Feature Films

Drug use has been a theme in the film industry for decades. Filmmakers often address both health-enhancing and risky behaviors, and at the same time, films with scenes of drug use have the potential to model behavior and convey normative propositions on issues such as unsafe sex and recreational drug use, among others [49].

Film productions with these themes began to appear as early as the fourth decade of the twentieth century, but at that time, they were more often concerned with alcohol dependence and the relationships shown between alcoholics and their relatives. In the last two decades of the twentieth century, many more films appeared in which drugs other than alcohol were used. Such scenes also began to form the main plot of screen adaptations [50]. Examples of films with frequent drug scenes include Trainspotting, Requiem for a Dream, Blue Collar, Easy rider, Big Lebowski, Drugstore cowboy, Bad Lieutenant, Traffic, Blow, Leaving Las Vegas, Kids, and many other film adaptations too numerous to list here [51,52]. The page ‘List of drug-related films’ can be found on Wikipedia, which contains an alphabetical list of several hundred films in which drug use is a significant theme or scenes of drug use or distribution. The drugs most frequently depicted in films are cocaine, heroin, cannabis, methamphetamines, and LSD [53].

Drug use is depicted in both films seen by international and national audiences, and film representations of drug use scenes may reflect the social image of drug use in a particular country, for example, Brazil [54], Scotland [31], Mexico, and Argentina [55]. In addition, many cinematic representations of drugs show a positive or neutral context of drug use, especially marijuana [7,49].

Castaldelli-Maia et al. conducted a study to assess the representation of drug use in scenes from Oscar-nominated films between 2008 and 2011 [56]. The authors analyzed the media content of 47 films (nominated for Best Picture, Best Actor, and Best Actress) depicting drug use and/or its consequences. The researchers identified a total of 515 scenes of drug use in these films. Both alcohol, cigarette, and illicit drug use were presented as problematic or occasional behaviors, but usually in response to stress and tension. Moreover, an increase was observed annually in scenes of drug use other than alcohol and tobacco and scenes of simultaneous use of multiple drugs, including by women. According to the authors of the study, films with episodes of drug use reflect what is happening in society; in Western countries, more and more teenagers experiment with drugs, e.g., cannabis, while tobacco smoking is declining. At the same time, the researchers point out that Oscar-nominated films are among the most popular films and may influence the creation of behaviors of the recipients of this content [56].

In these films and others that feature scenes of drug use, the context of use and the phenomenon itself are not only presented neutrally or positively. Numerous productions with bleak depictions of family, relational and social dysfunction, such as Requiem for a Dream, are considered cult films in this genre. However, even in this dramatic film, which has been described as a ‘horror film’, the scenes of drug use can imply a desire for a drugged experience. Short sequences that vividly depict close-ups of drug packages being opened, preparation, drugs being ingested, distribution in the bloodstream, and pupils dilating after use can create an exciting need for the viewer to experience these states [57].

In addition, many positive or neutral representations of tobacco, alcohol, and drug use can be seen in contemporary film productions [7,49,54,55]. In earlier film adaptations of drug use scenes, such as the aforementioned Requiem for a Dream or Trainspotting, the message was relatively unambiguous and could inform future physicians about drug behavior and medical methods [51]; however, in these productions, some scenes, due to the attractive depiction of the moment of drug intake and colorful experiences, can also arouse interest in this mysterious and interestingly depicted activity.

Movies and shows available on digital streaming platforms, including Netflix, Hulu, HBOMax, and others, are top-rated, such as the ‘Breaking Bad’ series depicting methamphetamine use, also play an essential role in disseminating content that can foster a change in drug perceptions. According to Brian Braiker, methamphetamine has never had better marketing than in the ‘Breaking Bad’ series [58].

In April 2021, the third edition of the SPLIFF Film Festival took place online. As the originators of this project state: “SPLIFF is a place where filmmakers, artists, animators, and smokers share original short films that explore and (or) celebrate cannabis and its liberating effects on our imagination, appetites, libido and creative energy” [59].

### 3.2. Music

Messages in the text layer of musical works, in which their authors often directly popularize drug use, may play a similar role in the previously described representations of drug use in films. At the end of the twentieth century, it was warned that the musical preferences of adolescents might correlate positively with engaging in risky behaviors, such as reaching for drugs [60]. Analogous observations were reported by members of the Council on Communications and Media [61] at the end of the new millennium’s first decade. Attention was paid primarily to representations of drug use in the lyrics of contemporary popular music.

In 2005, a content analysis of almost three hundred of the most popular songs (according to Billboard) was conducted to determine the frequency of drug-alcohol messages in the lyrics of selected music genres. The results of the study indicated that in pop songs, 9% of the content of this type was observed, in rock songs—14%, in hip-hop and R&B songs, 20% of the content about taking intoxicants was marked, and the highest indications were obtained in rap lyrics containing 77% of such messages. The authors of the study pointed out that music can be a reason for creating attitudes towards drugs. The measurement did not determine the impact of this type of content on youth behavior and focused solely on analyzing the text layer [62].

In addition, numerous other studies have observed associations of specific music genres (rap, reggae, techno, R&B, punk, heavy metal, house, trance) with alcohol use [27,63,64,65], and drug use, including cannabis, amphetamine, ecstasy, LSD, hallucinogenic mushrooms, heroin and GHB [27,63,65,66,67,68,69,70]. In addition, information on music performers’ CDs suggests that drugs are a fairly common theme, both visualized on the wrappers of the CDs and in their titles. Drugs are also sometimes cited as an inspiration for an album [70]. Studies have confirmed that the number of drug references in popular music lyrics has increased manifold in recent decades, while drug use is most often associated in these messages with splendor, wealth, mood enhancement, sexual activity, celebration, and social life [71,72,73].

Researchers on the relationship between music and behavior emphasize that even compositions without references to extra-musical reality can provoke strong psychological reactions in listeners [74], while texts transmitted through music can favor the popularization of the messages they contain [70]. Experiencing music is an experience of identity; the individual reacts to the piece being listened to and becomes involved in emotional alliances with the performers and other fans, creating a solid connection. The content heard can be absorbed and then adopted and brought into the listener’s life as their own [75].

### 3.3. Books, Specialist Magazines, Publications

Publications published in the form of books are also an example of promoting drug use and encouraging this kind of activity. Among titles worth mentioning are: *Food of the Gods* and *True Hallucinations* by Terence McKenna revealing mysterious effects of narcotics, works by LSD discoverer Albert Hoffmann and authors such as Timothy Leary, Richard Alpert, and Ralph Metzner describing their drug experiments, and Daniel Pinchbeck, who in his book Breaking the Mind encourages to change the perception of psychoactive substances and to use their properties [7].

Books with a decidedly liberal attitude towards drugs are also published by the Polish publishing house ‘Latawiec’, which has published, among others, A. Hoffmann’s work: *LSD… my difficult child* and a book entitled *Marihuana: the first twelve thousand years*, which is a study of this drug full of positive references. Similar items have been made available through other publishing houses distributing literature with a decidedly ambiguous message [7].

The dissemination of content saturated with positive information and messages about the use of psychoactive drugs is also exemplified by specialist magazines and publications aimed at those interested in psychoactive sensations, published by websites or available in paper form, including *Soft Secret, Spliff—the Hemp Newspaper*, which promotes the legalization of cannabis, and the attractively published magazine *Trans/visions—Psychoactive Journal* devoted to sensations after narcotics [7]. Obtaining such magazines is not difficult; they can be downloaded from websites, ordered by mail order, including archived issues, or encountered by chance as a promotional supplement to a book purchased from one of the publishing houses, as mentioned earlier.

The online platform feedspot.com updated in early April 2021, provides a ranking of 30 magazines publishing cannabis-related content to discuss cannabis news, companies, stocks, and technological advances, presenting the latest industry news, legal and financial information, business opportunities, cannabis compliance, among other topics. These publications also provide information on how to grow cannabis, strategies for growers and owners of finished product distribution businesses, current market data, and much more [76,77].

### 3.4. Celebrities

The glorification of drug use by celebrities and the prevalence of such behavior suggested by representatives of this group are further factors that may favor the liberalization of attitudes towards drugs [78]. Representatives of this group strongly influence the health behavior of their admirers, often turning them into followers of their preferred activities [79]. Projects have been undertaken in which researchers attempt to determine the impact of media coverage of idol drug use on adolescent behavior [80]. However, determining these relationships may not always be accurate as they may depend on the context in which the information is communicated; for example, reports of Amy Winehouse and Prince’s drug-related deaths may negatively influence perceptions of drug use [80,81].

Research suggests that watching scenes in which actors, especially celebrities, use tobacco, alcohol, and other drugs, or discuss their use, may influence viewers’ beliefs and behaviors regarding using these drugs. In a study among adolescents aged 12–15 years, it was found that smoking by film stars can play an essential role in encouraging adolescents, especially girls, to reproduce this behavior [82].

Sometimes depictions of psychoactive drug use are discussed and criticized by celebrities. However, sometimes the messages broadcast are inconsistent and create dissonance between the verbal and visual messages, as noted in analyses of a series depicting the life of rock star Ozzy Osbourne and his family [83]. Adolescents are a group particularly susceptible to such messages and quick to notice emerging differences and irregularities. At the same time, one has to consider irresponsible celebrities whose statements may pose serious health risks to their fans [84].

Studies conducted among adolescents have confirmed that despite the awareness of some pathological behaviors of their idols, adolescents unreflectively copy the activities observed in them. Along with the indifference and departure from previous authorities noticeable among adolescents, this phenomenon poses a considerable threat and challenge both to parents, teachers, and individuals or groups capable of influencing representatives of this age group. Celebrities admitting to using psychoactive substances may model such activity patterns among youth. For young people, idols are almost objects of worship, and at the same time, they are often mistakenly perceived as authorities. Unfortunately, they are often people who despise social norms, who have many problems and for whom using drugs is the norm, which they do not intend to hide [7].

### 3.5. Internet

Many researchers emphasize the vital role of social networking sites in the exchange of information among drug users. Many films, including on YouTube, depicting drug-related behavior can be found on the internet [85]. There are also portals on the internet created for everyone interested in drug use, e.g., *Erowid*, *Bluelight* [86], *HipForums* [87], *Hyperreal*, and *Neurogroove* [7]. In addition, popular social networks, e.g., *Facebook*, *Twitter*, and *Instagram*, are platforms often used for sharing drug experiences [88,89]. On these portals, it is possible to obtain all the information necessary for the home production of potent drugs produced from poppy seeds, over-the-counter (OTC) medicines combined with components available in every household, and information on all methods of achieving intoxication states [90,91]. Increasingly, drug traffickers are reaching out to young people through popular phone apps. In Denmark, Snapchat is the second (after Facebook) most popular virtual venue for drug trafficking. Dealers are ruthless and only care about getting new customers [92]. There are reports of deaths among even random teenage customers of drug dealers [93]. This knowledge can be obtained without any obstacles. As Paul M. Wax points out, it only takes one click to both obtain the information sought on drug use, its manufacture, ‘safe’ use and to purchase the goods sought via the internet [94].

The internet influences both the distribution and use of drugs; it facilitates the emergence of new producers and distributors in the global drug market while providing these actors with new customers and increasing demands for drugs [95].

In 2013, the EMCDDA identified 651 online sites offering new drugs, while in 2014, observation of emerging online offers confirmed the activity of portals offering other drugs for trade, previously only available on the so-called black market [96]. Acquisition of drugs is straightforward; after placing an order, a consignment is delivered to the indicated address within a few days, for which payment can be made only upon delivery. Traffickers often enhance standard consignments with free samples of new drugs. Attractive distribution and clear conditions set by online sellers create the appearance of legality, encouraging the use of the offered products [7].

In addition, there are quasi-pharmacies on the internet, where it is easy to purchase potent drugs without a prescription [97]. For more than a decade, an indeterminate number of shops offering Needle and Syringe Programs (NSP) and other drugs for trade have also been operating on the internet [98,99].

Internet surveys conducted among NSP users confirm that Internet forums are the primary source of information on new drugs. After drug intoxication incidents, people who go to treatment centers admit that they have used the information posted, among others, on the *Hyperreal* website [7]. In addition, knowledge of NSP can be obtained from internet shops, dealers, friends, and the media. Other studies have also found that the main reason for using NSP was to come across positive accounts of drug use on Internet portals [100]. Organized drug trafficking via social media, including Instagram and Facebook, advertising drugs through these media, and the downplaying of the risks associated with their use is a problem systematically observed by law enforcement and public health agencies [101,102].

### 3.6. Food

For more than a decade, the food market has seen products infused with cannabinoids present, including tetrahydrocannabinol (THC). These products come in various forms, such as baked goods, sweets, or beverages. Countries where cannabis use has been legalized have taken the necessary regulatory steps to reduce the risk of intoxication by requiring edible products to have universal warning symbols, informing consumers of the correct serving size; limiting the amount of THC per serving and the total number of servings per item [103]. Unfortunately, poisonings of cannabis-based food products have been reported, even ending in user death [104]. However, in countries where the use of cannabis derivatives has been legalized, products labeled with a cannabis leaf can be purchased in regular grocery shops to attract the attention of those interested in using the drug [7]. In early 2021, ‘BUH’ beer was advertised as a product containing hemp extract (CBD) was and was mass-marketed in Poland. Although the content of hemp derivatives in alcohol was questioned and the information given was a clever marketing effort, the product aroused considerable interest among customers, and the producers intend to expand distribution to domestic small and large food shops [105]. Food products containing hemp derivatives are sometimes referred to as ‘super foods’ [106], and producers encourage purchase and consumption by creating new trends that adolescents can assimilate. The observation of consumer behavior of adolescents confirms the high popularity of these types of food products [107].

### 3.7. Clothing, Jewellery, Gadgets, and Cosmetics

Recent decades have seen a dynamic development of the clothing industry, which targets its products to specific consumer groups, including psychoactive drug users. It should be noted here that research on the worn clothing as a stimulus to undertake specific behaviors and clothing as self-expression and identification with a given social group confirm the occurrence of dependencies and links between these variables [108].

Clothing manufacturers offer various models of shirts, trousers, caps, sweatshirts, coats, socks, and other garments, both full of pop culture references and, of course, full of weed leaf prints containing messages glorifying the use of cannabis derivatives. Observers of this phenomenon believe that it is a developing trend and unstoppable [109,110]. Brands are emerging whose activities are directed towards producing both clothing and all gadgets related to the use of cannabis: decorative boxes, pipes, blotters, grinders, and other smoking utensils [111,112,113]. In addition, entire series of drug-related jewelry collections are being created, where earrings, rings, necklaces, bracelets, and other intricately crafted items offered are in the shape of cannabis leaves or other drugs [109,114], or necklace pendants shaped like the molecular structures of cocaine, methamphetamine, LSD, and other drugs [115].

Cosmetics drugstores are increasingly selling products decorated with hemp leaves. Even though they do not contain psychoactive substances in their composition, due to the social discourse related to the controversy over the legalization of marijuana, these products are also popular among consumers [116,117].

## 4. Discussion

Mass media can both promote pro-health culture and foster anti-health activities [118]. Many studies suggest that adolescents’ risky behaviors are linked to exposure to drug-liberal mass culture content depicted in film, music, literature, websites, and everyday products: food, clothing, cosmetics, and gadgets. Due to the growing number of studies conducted in establishing relationships between drug representations in mass culture and attitudes towards using these drugs, we present only a few.

A study that aimed to establish links between smoking scenes in films and nicotine initiation among adolescents viewing these films (*n* = 6522) found that viewing such scenes can activate smoking, and this behavior can then be followed up and conditioned by other factors, in particular, the immediate social context of smoking (e.g., growing up/being around smokers) and consequently nicotine dependence. However, the authors point out that these data should not be generalized [119].

A study that examined the influence of scenes of psychoactive drug use observed in the media on the use of such specifics found associations between these variables: respondents admitted that viewing mass media content with drug messages influenced their decision to use these drugs [40].

When analyzing posts available on Instagram about opioid abuse, the researchers observed that codeine misuse becomes commercialized and ritualized. Furthermore, these posts instruct and normalize this type of drug abuse while linking codeine use to pop culture icons (The Simpsons, Mickey Mouse, or Pokemon) in posts by appealing to humorous messages concerning dangers. At the same time, a frequent topic of discussion on Instagram was combining codeine with cannabis, alcohol, and benzodiazepines, which significantly increases the risk of consequences, including overdose mortality, especially among adolescents [89].

A 2011 study of US adolescents found that adolescents who regularly spend time on social networking sites are more likely to engage in risky behaviors, including smoking, alcohol consumption, and marijuana use. Researchers have attributed an essential role to social networking sites, including Facebook, Myspace, and others, where adolescents view photos of other users of the sites using psychoactive substances or view content on such sites that illustrate states experienced after drug use [120].

A study conducted among Polish adolescents (*n* = 2273) found statistically significant associations of listening to music, watching films, and browsing websites with liberal drug messages on attitudes towards drugs and increased indications of use reported by respondents [7].

The cited research data correspond to the cultivation theory of Gerbner and L. Gross [41], and the results of these studies suggest that the behavioral patterns shown in the media are perpetuated and nurtured, and transferred in the actions of the audience in the social space in which they function, learn, work, play, and live.

The ease of assimilating positions created by the lyrics of musical pieces or content present on Internet portals is strengthened by mutual interactions occurring, for example, during large-scale youth music shows. Therefore, these results correspond to the assumptions of A. Bandura, indicating that drug use may be the result of taking over behavior patterns from persons significant for the individual [45].

The observed relations between the use of popular culture products and drug use also correspond to the features of social reality defined by the postmodern perspective: ambivalence, complete tolerance of all ideas, pluralism, liberalization, the aftermath of “modernity” defined by J. Baudrillard as “the state after the orgy”—the emergence of liberation in all possible spheres of social reality, including political liberation, sexual liberation, the liberation of women, children, unconscious drives, destructive powers, productive forces, and at the same time the affirmation of all models of representation and anti-representation [121].

However, to confirm both Baudrillard’s suppositions and the considerations of the authors of this article, it is necessary to undertake well-designed research to precisely define these relational relationships [122].

## 5. Conclusions

The above data suggest an essential role of popular culture products (especially messages in contemporary music lyrics and content available on websites) in creating liberal, pro-drug positions. Other contents of popular culture which may influence drug use (e.g., films and books presenting drugs in a positive light and pro-drug declarations of famous and popular people admitting their experiences with drugs) are also of considerable importance.

The observations presented in this article consider only some of the many research issues related to this socially momentous phenomenon. The topic itself is a multidimensional, multifaceted problem. The material presented in this paper can be successfully used by practitioners (sociologists, pedagogues, and psychologists) and institutions focused on social prevention. It can also be a starting point for researching this critical social phenomenon. Suggested actions include:−monitoring the representation of drug use in mass culture content;−conducting reliable research on the impact of mass culture content on audiences;−identify possible causal links between the use of such content and changes in attitudes towards and use of drugs;−introducing preventive measures, including education on the critical reception of mass culture content.

## Data Availability

Not applicable.

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
