# Peer review of "Representations of Psychoactive Drugs’ Use in Mass Culture and Their Impact on Audiences"

_ijerph, 2021, doi:10.3390/ijerph18116000_

Round 1
Reviewer 1 Report
I read the review by Marek A. Motyka and Ahmed Al-Imam with much interest. The authors presented a well-written review of drug representations in the content of mass culture, e.g., in film, music, literature, and the occurrence of drug references in everyday products, e.g., food, clothes, cosmetics. And how it may influence the drug use.
Overall, this is a balanced, comprehensive review of drug representations in mass media and its impact on the audience.
Author Response
We are very grateful for recognising our work as meeting the scientific requirements and contributing to science.
Reviewer 2 Report
Dear Authors,
Nice, topical and an interesting review. However, I have some minor objections addressed to the manuscript:
1) please, add also snus usage as one of the serious problem nowadays in 1-2 sentences, and critically analyse in one paragraph the circulating in the society idea that some psychoactive drug (snus stimulator, marijuana) are not do "dangerous" as the other ones...
2) correct the Conclusions - remove please the personalizations, references (that paragraph might be moved to the main text, by the way), extra words, - make them more short and precise!
3) check once more the References, - are you sure that you really need these 5 old previous century sources? They do not fit nicely for this manuscript, - perhaps you can remove them or replace with some more latest ones.
Author Response
Thank you very much for your positive review and valuable comments. We have followed the reviewer's comments: added a reference to snus, inserted a critical comment, corrected the Conclusions section, and updated the References. We left one footnote to the late 20th-century article [66] because it relates to the historical context of the description. We ask you to include this reference; in our opinion, it is essential.
Reviewer 3 Report
The article is well-written and focused on describing the effects of mass culture on psychoactive drug use on audiences.
Minor comments:
- The introduction (first paragraph, page 1) describes the amount of drug users from 2009-2018, is there also information about the number of drug overdose leading to death?
- In two parts of the manuscript (Page 1 - line 34, and Page 2 - line 52), the text says: "...is particularly important to introduce prevention strategies different from the ones used so far...", which are these prevention strategies?
- In the section Feature Films, the manuscript should also mention TV Series in Netflix, Hulu, HBOMax, etc., which have been very popular. And example is the series "Breaking Bad".
- In the section Internet, MySpace is no longer a popular social network. In addition, the section could mention the use of application or App's, such as Snapchat, which recently has been associated with the selling and buying of drugs in adolescents. There are news article about it on Fox4: https://fox4kc.com/news/teens-deadly-overdose-prompts-warning-about-snapchats-role-in-buying-selling-drugs/ and https://www.businessinsider.com/snapchat-instagram-drug-dealers-fentanyl-counterfeit-pills-teen-deaths-2021-3
Author Response
The authors would like to thank you for your positive review and valuable comments; thanks to them, we learned about significant issues. In the text, we have made corrections in all the indicated places:
- we added information on deaths (from both reports);
- we pointed out examples of strategies that need to be implemented;
- we added information about streaming platforms;
- we removed the entry about MySpace and added information about drug trafficking on Snapchat.